# Restorative Materials Exposed to Acid Challenge: Influence of Temperature on In Vitro Weight Loss

**DOI:** 10.3390/biomimetics7010030

**Published:** 2022-02-26

**Authors:** Riccardo Beltrami, Marco Colombo, Gianpaolo Bitonti, Marco Chiesa, Claudio Poggio, Giampiero Pietrocola

**Affiliations:** 1Section of Dentistry, Department of Clinical-Surgical, Diagnostic and Pediatric Sciences, University of Pavia, 27100 Pavia, Italy; marco.colombo@unipv.it (M.C.); gianpaolo.bitonti01@universitadipavia.it (G.B.); marco.chiesa@unipv.it (M.C.); claudio.poggio@unipv.it (C.P.); 2Department of Molecular Medicine, Unit of Biochemistry, University of Pavia, 27100 Pavia, Italy; giampiero.pietrocola@unipv.it

**Keywords:** restorative dentistry, resin composite materials, glass-ionomer cements, acid exposure, temperature

## Abstract

Consumption of acidic beverages and foods could provoke erosive damage, both for teeth and for restorative materials. Temperatures of consumption could influence the erosive effects of these products. The aim of this in vitro study is to assess the influence of an acidic challenge on the weight loss of different restorative materials. Resin composites and glass-ionomer cements (GIC) were tested. The medium of storage was Coca-Cola (Coca-Cola, Coca-Cola Company, Milano, Italy) at two different temperatures, 4 and 37 °C, respectively for Group A and Group B. For each group, nine specimens were prepared for each material tested. After 7 days, weight was assessed for each sample, and the percentage weight loss was calculated. For all the resin composites (Groups 1–13), no significant weight losses were noticed. (<1%). Conversely, GICs (Groups 14 and 15) showed significant weight loss during the acidic challenge, which was reduced in the case of these materials that included a protective layer applied above. Significant differences were registered with intra-group analysis; weight loss for specimens immersed in Coca Cola at 37 °C was significantly higher for almost all materials tested when compared to specimens exposed to a cooler medium. In conclusion, all the resin composites showed reliable behaviour when exposed to acidic erosion, whereas glass-ionomer cements generally tended to solubilize.

## 1. Introduction

A tooth may require restoration or filling for many reasons [1,2,3,4,5,6,7]. The most frequent reasons are caries, erosion, trauma, abrasion, congenital anomalies and aesthetically defective teeth. Among these, erosion seems to be one of the most prevalent reasons for intervention. The chemical loss or softening of enamel and dentin is not produced by bacteria but is due to the action of acids [8]. Aggressive erosion or enamel softening are also typical signs in patients with psychosocial pathological disorder, such as anorexia and bulimia, or autoimmune syndromes, such as Sjogren’s syndrome [9]. Sjogren’s syndrome causes a significant reduction in saliva, which in physiological conditions, provides protection through dilution, buffering, neutralization and elimination of acids, as well as through providing minerals for the remineralization of the eroded surface [10]. Except for this particular case, diet represents the key aetiology factor, and particular attention has been focused on acidic drinks [11,12,13]. Acidic solutions may destroy the hard tissue of the teeth by erosion if prolonged contact occurs [14]. Consumption of nonalcoholic beverages such as soft drinks is therefore the main cause of dental erosion in young patients [15,16,17,18,19,20,21]. Several studies have described the significant loss of hardness on enamel after immersion in different soft drinks such as orange juice, fruit juices or Coca-Cola, but the range of studies published on this topic could be easily widened if the research would comprise citric acid or hydrochloric acid. Citric acid is commonly chosen as a substitute for acidic beverages and is classified as a medium-strong to weak polyvalent acid. Hydrochloric acid was chosen because it is a compound of gastric juice. In vitro published research has shown that citric acid, when compared to hydrochloric acid, exhibits a great potential for further dissociation and delivery of H^+^ protons. Moreover, it is reported that citric acid has chelating properties for enamel’s calcium ions, thus enhancing further dissolution. Therefore, the acid erosion of enamel comprises different chemical mechanisms induced by the acid solution: type of acid, pH, amount of tithable acid, and chelating ability, in which calcium and phosphate have total impact on the degree of erosivity of any beverage. Even physical aspects seem to have influence on the erosive potential of acidic beverages; however, until now, few research regarding adhesiveness of the solution to the enamel surface, agitation and flow of the beverage, and temperature of consumption has been published [22,23,24]. These interesting theoretical results could be shifted to clinical situations adopting acidic beverages as a medium for enamel specimens [25]. Grobler et al. [26] reported that fruit juices are much more destructive to teeth than whole fruit due to the percentages and quality of sugar contained [27]. Gedalia et al. [28] stated that the microhardness of enamel decreased in only an hour from exposure to Coca-Cola due to the fact that it contains phosphoric acid, which is considered extremely erosive. In addition, soft drinks are generally consumed at a considerably different temperature from equilibrium temperature of the mouth at 36 °C. The consumption of hot or cold fluids causes a change in oral temperature as previously described by Airoldi et al. [29]. Chemical reactions, such as the dissociation of acidic substances, are thermodynamically favoured if environmental temperature is higher [30]. Therefore, the temperature of consumption of acidic drinks could affect the erosive lesion depths in a significant manner.

Aesthetic restorative composite resins should maintain the appearance of natural teeth, but the durability in the mouth is related to their microhardness and insolubility. Therefore, the erosive acidity of soft drinks introduced with diet affects the microhardness, wear and microleakage of the composite resins and the durability of the dental restoration in the long term [31,32,33]. Coehlo et al. [34] reported that generally there is a surface decrease in microhardness as well as an increase in roughness in the long term. The subsequent weight loss of the dental restorations could be measured in an in vitro design study and could be related to the acidogenic potential of the soft drinks. Clinicians could apply different strategies to minimize the effects of weight loss of dental restorations, such as informing patients about durability of restorations, advising patients regarding alimentary habits and use of topically applied fluoride formulations, performing preventive and minimally invasive treatments when the decay of restorations is evident [9].

A wide range of materials by different manufacturers has been tested, particularly composite resins and glass-ionomer cements. Composite resins allow one to carry out permanent dental restorations with good aesthetic and adequate mechanical/chemical parameters; glass-ionomer cements are more frequently used in orthodontics and pedodontics.

Physiological and para-physiological conditions could therefore change mechanical and chemical characteristics of restorative materials. Besides, several research articles have assessed, in vitro, different aspects and have shown a significant general decrease in surface microhardness as well as an increase in roughness [34]. The storage medium for the materials tested is therefore prepared mimicking the challenging situations, such as the case of high intake of acidic drinks.

The aim of the present study is to evaluate and compare the action of acidic challenges on the weight loss of restorative composite resins and of two glass-ionomer cements from different manufacturers. The tested hypothesis is that acidic drinks and their temperature of consumption cause no erosion, and consequently weight loss, of the restorative materials tested.

## 2. Materials and Methods

### 2.1. Materials Tested

Thirteen different composite resins and two glass-ionomer cements (GICs) were considered in this study and subdivided into groups. Table 1 reports the compositions and manufacturers of each material testes.

### 2.2. Determination of Sample Size

To determine a valid sample size (alpha = 0.05; power = 80%), we hypothesized an expected mean for percentage weight loss of 1.75 with a standard deviation of 0.85. The expected difference between the means was supposed to be 1.35, and therefore 9 specimens were requested for each group. For each material, we prepared 18 specimens as described above. The specimens were then randomly divided into two groups.

### 2.3. Sample Preparation

In order to obtain equal specimens, silicon rings (height 2 mm, internal diameter 6 mm, external diameter 8 mm) were filled with the materials tested. Each sample of glass-ionomer cements was prepared by mixing liquid and powder, and then silicon rings were filled as described. For each sample of glass-ionomer cements, we mixed powder and liquid, in order to fill molds with freshly prepared material. Molds were positioned above a dark opaque paper background with a polyester matrix strip interposed in order to obtain a smooth surface under the material, as well as to avoid light reflection from the bottom, thus reducing artificial hardening of this area. For each product, the A2 Vita shade was chosen in order to avoid the effects of colorants on light curing [35].

Each mould was slightly overfilled, and a second polyester matrix strip (Mylar strip, Henry Schein, Melville, NY, USA) was positioned on the top to avoid oxygen interfering with the polymerization of the most superficial layer of the composite; in order to extrude the excess composite resin and obtain a flat surface, a glass slide was pressed against the upper polyester film and removed before curing [36].

Each sample of the light-curing composite resins was light cured for 40 s with the LED unit Celalux 2 (Voco, Cuxhaven, Germany) and then removed from the mold without conducting polishing. Led unit Celalux 2 was checked with a radiometer (SDS Kerr, Orange, CA, USA) before every use. The terminal device of the LED unit was placed on the external (top) side of the molds and concentrically with the rings. Exclusively, one light polymerization mode was used, with an output irradiance of 1000 mW/cm^2^ [37,38].

Each sample of the glass-ionomer material was allowed to harden for the same time as reported in manufacturer’s instructions.

All the samples were subsequently dried at 37 °C for 24 h and then weighed with a Mettler–Toledo precision balance (AE1633, Mettler-Toledo SPA, Novate Milanese, Milan, Italy) with metering accuracy of 0.01 mg. Three weights were registered for each specimen, and mean value was considered for analysis.

Subsequently, 18 specimens of each material were divided into two experimental groups:Group A: nine specimens immersed in 50 mL of a soft drink (Coca-Cola, Coca-Cola Company, Milano, Italy) at temperature 4 ± 1 °C;Group B: nine specimens immersed in 50 mL of a soft drink (Coca-Cola, Coca-Cola Company, Milano, Italy) at temperature 37 ± 1 °C.

After 7 days, each specimen was removed from the liquid using tweezers, then dried with blotting paper, left undisturbed for 24 h at 37 °C to completely dry, and then weighed three times with the precision balance as previously described. Mean value for each specimen was considered for the analysis.

The difference between the mean weight before the immersion and the mean weight after the immersion was considered as the outcome of the study (WL: weight loss) and then expressed in percentage. The difference in percentage was considered the percentage of weight loss due to acidic erosion.

Weight loss (WL) = mean weight before immersion (Mw_t0_) − mean weight after immersion (Mw_t1_).

### 2.4. Statistical Analysis

Data were analysed with R (The R Foundation for Statistical Computing, Vienna, Austria). The resulting data were expressed as percentages of weight loss of materials after 7 days of acid challenge at different temperatures and pH. Descriptive statistic values median, minimum, maximum, mean and standard deviation were calculated. Data observed were not normally distributed, and non-parametric statistical methods were used for statistical analysis. The Wilcoxon test was used for intragroup comparison of weight loss of each material due to the different acidic expositions. The Kruskal–Wallis was used for intergroup multiple comparisons of the different materials tested. Significance value was set as *p* < 0.05.

## 3. Results

As shown in Table 2 and Figure 1, the exposure to the acidic drink of all restorative materials tested caused different values of weight loss. G-aenial, Enamel Plus Hai Bio Function, Grandioso Heavy Flow, Admira Fusion x-base, GrandiO Flow, G-aenial Flow, Ceram.x Spectra ST flow, Admira Fusion x-Tra, VisCalor Bulk, GC Equia Forte and GC Equia Forte + Coat assigned to Group A lost significantly less weight than Group B (*p* < 0.05), while Grandioso Flow showed the opposite result (*p* < 0.05). The protocols of acid exposure (Groups A and B) did not significantly affect the weight loss of Grandioso Light Flow, x-Tra Fil and GrandiOSO x-tra (*p* > 0.05), which showed similar results even if the acidic expositions at different temperatures were different. With intergroup multiple comparisons, it emerged that Enamel Plus Hai Bio Function, Admira Fusion x-base, G-aenial Flow, Ceram.x Spectra ST flow and Admira Fusion x-Tra had similar weight loss (*p* > 0.05) in Group A conditions, while G-aenial, Grandioso Light Flow, Grandioso Heavy Flow and x-Tra Fil had a significantly higher percentage of weight loss (*p* < 0.05). GC Equia Forte and GC Equia Forte + Coat showed similar results after exposure to conditions of Group A (*p* > 0.05). When considering materials exposed to Group B acidic conditions, intragroup multiple comparisons showed that GC Equia Forte and GC Equia Forte + Coat had the highest values of weight loss if compared to composite resins. Similar percentage values were recorded for G-aenial, GrandiOSO Light Flow, GrandiOSO Flow, x-tra Fill, G-aenial Flo-X, Ceram.x Spectra ST flow, Admira Fusion x.tra and GrandiOSO x-tra (*p* > 0.05). Admira Fusion x-base, GrandiO Flow and VisCalor Bulk showed similar low percentages of weight loss when exposed to conditions of Group B (*p* > 0.05).

## 4. Discussion

Currently, the large consumption of artificially sweetened beverages, sport drinks, energy drinks and other substances causes problems to restorative dental materials [39,40,41]. The reason is that the beverages come into contact with the oral cavity, and thus the pH of the beverage affects the cavity. Teeth undergo dental erosion due to the concentration of hydrogen ions, i.e., pH measurement. The softening and dissolution of dental tooth structure is primarily caused by weak acids such as citric and phosphoric acid. In the present study, to test the durability of restorative dental materials, Coca-Cola was selected because it is a frequently consumed beverage and because its pH is about 2.3–2.5. When the oral pH range is between 2.0 and 4.0, tooth enamel erodes, although enamel demineralization starts at a pH of less than 5.5 [42].

The composition of each restorative dental material and polishing techniques have a direct impact on final surface characteristics of the restorations. Roughness, hardness, susceptibility to erosion and susceptibility to staining could then be influenced by environmental factors [43]. In this study, specimens of all materials tested were polymerized under a polyester matrix strip because they are reported to give the smoothest surface in experimental studies [44].

It can be seen from Table 2 that a higher temperature of soft drink could cause a higher weight loss in almost all materials tested. The differences between the mean percentages are significant. An increase in temperature of about 33 °C causes an average decrease of 0.15–0.20 pH units for the beverage [45,46]. Grandioso Light Flow, x-Tra Fill and Grandioso x-Tra were the only restorative materials of which an increase in weight loss due to higher temperature was registered, but it was not statistically significant.

Several authors have found that restorative materials subjected to thermal changes in the oral environment undergo unfavourable effects on the margins of the restorations, thus provoking microleakage and secondary caries [44,47]. In vivo studies reported that the salivary buffering capacity could mitigate or increase the erosive effects of acidic beverages on enamel and on restorative dental materials. Sanchez et al. reported that low salivary flow rates are associated with wider eroded areas on enamel [47]. However, the influence of this confounder is unpredictable and is not reproducible in experimental assays [47]. In the literature, there is confusion about erosion characteristics and solubility of dental restorative materials between clinical and in vitro studies [48,49,50]. The application of different experimental protocols and media affects the possibility to fully understand and compare the physico-chemical properties of different restorative materials when in direct contact with acidic solutions.

The present study shows that all the composite resins tested have proven to be well resistant to the acid medium. Conversely, some glass-ionomer cements can be subjected to an elevated and progressive loss of weight after exposure to acidic beverages, but if they are coated with a protective material, this loss significantly decreases. The mechanism of loss of weight in acid buffer solutions depends on the diffusion of the eluted species in the function of H^+^ ion concentration. The dissolution of the cement is therefore the result of the diffusion of ion concentration and the surface reaction. The application of the coat in GC Equia Forte + Coat is motivated by these chemical aspects that significantly influence the behaviour of glass-ionomer cements [51]. Above these considerations, glass-ionomer cements remain among the most used materials in paediatric dentistry and orthodontics because of their action as a fluoride reservoir which increases the concentration of this ion in saliva, plaque, and hard tissues of teeth, thus reducing the incidence of secondary caries [47].

These results suggest that restorative materials could be effective in protecting from erosive damages caused by excessive consumption of acidic drinks. Even for long immersion times, all materials tested did not lose an amount of weight higher than 0.3%, except for GrandiOSO Flow, which reached 1% of weight loss.

Our results show that there is an influence of acidic beverages and their temperature of consumption on the erosion of restorative materials. The mean weight loss of restorative materials for an acid solution at 4 °C was 0.3477, while for an acid solution at 37 °C, it was 0.5024. The entity of weight loss was minimal in percentage for composite resins (<1%), while significant weight loss was recorded for glass-ionomer cements.

The erosion kinetics, considering the dissolution steps of surface wash-off, the surface corrosion and the diffusion in the solid state, are not taken into account in the present study, and this could be addressed as the main limitation of this in vitro research. However, as reported by Matsuya et al. [52], the chemical kinetics undergoing the dissolution in acidic solution are well known and could be synthesized in two processes: the diffusion and the surface reaction between the acid anion and the eluted ions.

## 5. Conclusions

Further in vivo studies are needed to confirm our preliminary results; however, the methodology about the type of acidic solution, immersion time and polishing technique should be maintained to achieve comparable results. The limit of the present study is that environmental confounders, such as salivary buffering capacity and oral hygiene procedures, are not weighted in the in vitro analysis. In in vitro analyses, the differences in methodology could bring differences in the results, as shown in many studies published on this topic [33,53,54]. Moreover, the present study did not consider a control medium such as distilled water, which could have highlighted or reduced the effects of the acidic beverage.

## Figures and Tables

**Figure 1 biomimetics-07-00030-f001:**
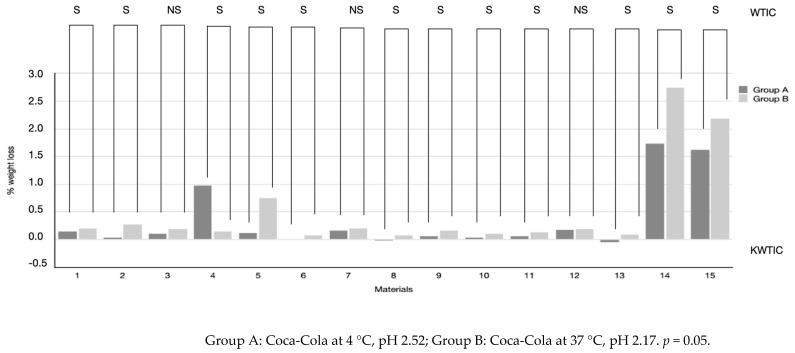
Representation of weight loss percentages after acid immersion in Coca-Cola at 4 °C (Group A) and Coca-Cola at 37 °C (Group B) for restorative materials tested. Wilcoxon test results for intragroup significance (WTIC) are reported in the rows for each material, and they evaluate the differences between Group A and B for each material (S: significant; NS: not significant).

**Table 1 biomimetics-07-00030-t001:** Restorative dental materials tested in the study.

Group	Material	Type	Composition	Filler % (*w*/*w*)	Manufacturer	Lot Number
1	G-ænial (Anterior)	Radiopaque Composite	Matrix: UDMA, dimethacrylate co-monomers, no bis-GMa Filler: silica, strontium, lanthanoid fluoride	76%	GC Corporation, Tokyo, Japan	1906251
2	Enamel plus Hri Bio Function	Microfilled hybrid composite	Matrix: UDMA, Tricyclodecane dimethanol dimethacrylate Filler: silicon dioxide	74%	Micerium S.p.A., Avegno, Italy	2018006379
3	GrandiOSO Light Flow	Flowable nanohybrid composite	Matrix: methacrylates (Bis-Gma, Bis-Ema, TEGDMA, 1,6 hexanodiylbismethacrylate, HEDMA). Filler: inorganic filler	76%	VOCO GmbH, Cuxhaven, Germany	1944439
4	GrandiOSO Flow	Flowable nanohybrid composite	Matrix: methacrylate (Bis-Gma, Bis-Ema, TEGDMA and HEDMA). Filler: inorganic fillers	81%	VOCO GmbH, Cuxhaven, Germany	1945398
5	GrandiOSO HeavyFlow	Flowable nanohybrid composite	Matrix: methacrylate (Bis-Gma, Bis-Ema, TEGDMA and HEDMA) Filler: inorganic fillers	83%	VOCO GmbH, Cuxhaven, Germany	1947547
6	Admira Fusion x-Base	Nanohybrid ceramic based composite	Matrix: ORMOCER^®^ Filler: glass ceramic, silica nanoparticles, pigments	72%	VOCO GmbH, Cuxhaven, Germany	1946562
7	x-Tra Fil	Hybrid composite	Matrix: methacrylate (Bis-GMA, UDMA, TEGDMA) Filler: inorganic filler	86%	VOCO GmbH, Cuxhaven, Germany	1946276
8	GrandiO Flow	Flowable nanohybrid composite	Matrix: methacrylate (Bis-Gma, Bis-Ema, TEGDMA and HEDMA) Filler: inorganic filler	80%	VOCO GmbH, Cuxhaven, Germany	1944463
9	G-ænial Flo X	Radiopaque Flowable composite	Matrix: UDMA, Bis-MPEPP, TEGDMA Filler: silicon dioxide, strontium glass	71%	GC Corporation, Tokyo, Japan	1905081
10	Ceram.x Spectra ST flow	Hybrid aesthetic composite	Matrix: BisGma adduct modified with urethane, BisEMA and diluents, stabilizers, pigments, camphorquinone photoinitiator Filler: based on Sphere TeC^®^ system	62.50%	Dentsply Sirona, Konstanz, Germany	1902000743
11	Admira Fusion x-Tra	Nanohybrid ORMOCER^®^ bulkfill composite	Matrix: ORMOCER^®^ Filler: glass ceramic, silica nanoparticles, pigments	84%	VOCO GmbH, Cuxhaven, Germany	1941488
12	GrandiOSO x-Tra	Nanohybrid bulkfill composite	Matrix: Bis-GMA, Bis- EMA, aliphatic dimethacrylate Filler: inorganic filler, organically modified silica	86%	VOCO GmbH, Cuxhaven, Germany	1938102
13	VisCalor Bulk	Thermoviscous nanohybrid bulkfill composite	Matrix: Bis-GMA, aliphatic dimethacrylate. Filler: inorganic filler	83%	VOCO GmbH, Cuxhaven, Germany	1946611
14	GC Equia Forte	Bulk Fill Glass Hybrid	Powder: fluoro-alumino-silicate glass, polyacrylic acid powder, pigment Liquid: polyacrylic acid, distilled water, polybasic carboxylic acid	/	GC Corporation, Tokyo, Japan	161020A
15	GC Equia Forte + Coat	Bulk Fill Glass Hybrid	Powder: fluoro-alumino-silicate glass, polyacrylic acid powder, pigment Liquid: polyacrylic acid, distilled water, polybasic carboxylic acid Light curing protective coating	/	GC Corporation, Tokyo, Japan	161020A Coat 1605131

**Table 2 biomimetics-07-00030-t002:** Weight loss of materials after 7 days of acid challenge. Data are expressed as medium percentage of weight loss (SD). Different capital letters indicate statistically significant Kruskal–Wallis results for intergroup differences (KWTIC) between materials tested. Wilcoxon test results for intragroup significance (WTIC) are reported in the rows for each material, and they evaluate the differences between Group A and B for each material (S: significant; NS: not significant).

Group A	7 Days	KWTIC	Group B	7 Days	KWTIC	WTIC
1A. G—aenial	0.149 (0.011)	A	1B. G—aenial	0.201 (0.049)	A	S
2A. Enamel plus HRi Bio Function	0.041 (0.035)	B	2B. Enamel plus Hri Bio Function	0.271 (0.010)	B	S
3A. GrandiOSO Light Flow	0.109 (0.139)	A	3B. GrandiOSO Light Flow	0.192 (0.031)	A	NS
4A. GrandiOSO Flow	0.977 (0.016)	C	4B. GrandiOSO Flow	0.150 (0.021)	A	S
5A. GrandiOSO HeavyFlow	0.125 (1.419)	A	5B. GrandiOSO HeavyFlow	0.752 (0.009)	C	S
6A. Admira Fusion x-base	0.0145 (0.015)	B	6B. Admira Fusion x-base	0.074 (0.015)	D	S
7A. x-Tra Fil	0.163 (0.034)	A	7B. x-Tra Fil	0.198 (0.010)	A	NS
8A. GrandiO Flow	−0.019 (0.017)	D	8B. GrandiO Flow	0.076 (0.043)	D	S
9A. G-aenial Flo-X	0.066 (0.016)	B	9B. G-aenial Flo-X	0.156 (0.024)	A	S
10A. Ceram.x Spectra ST flow	0.034 (0.015)	B	10B. Ceram.x Spectra ST flow	0.109 (0.019)	A	S
11A. Admira Fusion x-Tra	0.068 (0.031)	B	11B. Admira Fusion x-Tra	0.139 (0.013)	A	S
12A. GrandiOSO x-Tra	0.182 (0.009)	E	12B. GrandiOSO x-Tra	0.189 (0.011)	A	NS
13A. VisCalor Bulk	−0.044 (0.014)	F	13B. VisCalor Bulk	0.089 (0.013)	D	S
14A. GC Equia Forte	1.73 (0.15)	G	14B. GC Equia Forte	2.75 (0.006)	E	S
15A. GC Equia Forte + Coat	1.62 (0.14)	G	15B. GC Equia Forte + Coat	2.19 (0.35)	F	S

## Data Availability

Not applicable.

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
