# Peer review of "Restorative Materials Exposed to Acid Challenge: Influence of Temperature on In Vitro Weight Loss"

_biomimetics, 2022, doi:10.3390/biomimetics7010030_

Round 1

Reviewer 1 Report

This in vitro study aimed to investigate the influence of acidic challenges on the weight loss of various restorative materials. The background information given in the Introduction should be expanded by including measures to protect dental hard tissues and restorative materials with surface sealants. Furthermore, the erosive potential of acids should be explained in more detail, and the potential of other substances than acids regarding surface alterations of teeth and restorative materials should be mentioned. The following references might be helpful and cited: Acta Odontol Scand 2013, 71:1188-1194. doi: 10.3109/00016357.2012.757361; Arch Oral Biol 2020, 112:104686. doi: 10.1016/j.archoralbio.2020.104686. The results of the paper are unexpected. Usually, restorative materials gain weight after storage in liquids due to water uptake. How do the authors explain the weight loss of the materials and how was the calculation performed (the equation is missing)?

Author Response

Dear Reviewer,

thank you for tour precious suggestions.

We tried to improve the quality of our work following your advices.

You find our detailed answer in the file attached and in the manuscript.

Best regards

Riccardo Beltrami

Reviewer 2 Report

This article, titled” Restorative materials exposed to acid challenge: influence of temperature on in vitro weight loss” was conducted to be reviewed in Biomimetics. The manuscript is well written, but some specific issues still need to be clarified.

Specific

  1. How many samples are there in total? The number of samples in each group should be clearly stated in the abstract.
  2. Table 1: Is there an association between filler percentage and the study design?
  3. References are too old; please update.
  4. Figure 1, please mark the statistical differences.
  5. The authors should add discussion on the possible reasons why the weight loss rate of GIC (group 14, 15) is so different from other materials.
  6. Please follow the"MDPI : biomimetics" author guide. Some information, author contributions, funding, etc., should be listed.

Author Response

Dear Reviewer,

thank you for your precious suggestions.

We tried to improve the quality of our work following your advices.

You find our detailed answer in the file attached and in the manuscript.

Best regards

Riccardo Beltrami

Round 2

Reviewer 1 Report

Unfortunately, the authors were not able to significantly improve the manuscript. Their explanation for the unexpected weight loss is not convincing. The authors should study the literature on water sorption and solubility, and discuss their results in the context of findings of other papers. The authors also did not provide information on their calculations (the equation is still missing in the manuscript). Finally, the Introduction has only marginally improved. The erosive potential of acids should be explained in more detail (see for example Clin Oral Investig 2013, 17:595-600. doi: 10.1007/s00784-012-0731-3 as a helpful and relevant paper) and the potential of other substances than acids regarding surface alterations of teeth and restorative materials should be mentioned (see for example J Am Dent Assoc 2012, 143:1324-31. doi: 10.14219/jada.archive.2012.0095 as a helpful and relevant paper). The authors should revise their manuscript thoroughly to improve its quality.

Author Response

Dear Reviewer,

Revisions to the manuscript are marked in the text using the “Track Changes” function. Here in the cover letter we provided a point by point response to the reviewer’s comments. We appreciate your help in improving our experimental research.

We hope we could get in touch with you, or viceversa, to plan together new interesting projects regarding dental materials.

Thank you for your suggestion

Best regards

Riccardo Beltrami
